# A dual-branch model combining convolution and vision transformer for crop disease classification

Qingduan Meng◉, Jiadong Guo◉, Hui Zhang◉, Yaoqi Zhou◉, Xiaoling Zhang◉*

College of Information Engineering, Henan University of Science and Technology, Luoyang, Henan, China

◉ These authors contributed equally to this work.
* sunnyzhangxl@163.com

## Abstract

Computer vision holds tremendous potential in crop disease classification, but the complex texture and shape characteristics of crop diseases make disease classification challenging. To address these issues, this paper proposes a dual-branch model for crop disease classification, which combines Convolutional Neural Network (CNN) with Vision Transformer (ViT). Here, the convolutional branch is utilized to capture the local features while the Transformer branch is utilized to handle global features. A learnable parameter is used to achieve a linear weighted fusion of these two types of features. An Aggregated Local Perceptive Feed Forward Layer (ALP-FFN) is introduced to enhance the model's representation capability by introducing locality into the Transformer encoder. Furthermore, this paper constructs a lightweight Transformer block using ALP-FFN and a linear self-attention mechanism to reduce the model's parameters and computational cost. The proposed model achieves an exceptional classification accuracy of 99.71% on the Plant-Village dataset with only 4.9M parameters and 0.62G FLOPs, surpassing the state-of-the-art TNT-S model (accuracy: 99.11%, parameters: 23.31M, FLOPs: 4.85G) by 0.6%. On the Potato Leaf dataset, the model attains 98.78% classification accuracy, outperforming the advanced ResNet-18 model (accuracy: 98.05%, parameters: 11.18M, FLOPs: 1.82G) by 0.73%. The model proposed in this paper effectively combines the advantages of CNN and ViT while maintaining a lightweight design, providing an effective method for the precise identification of crop diseases.

## 1. Introduction

Agriculture is a crucial cornerstone of human survival. However, disease infections in crops profoundly disrupt the normal physiological process of the crops, resulting in yield reduction, quality degradation, and even crop failure. Statistics indicate that agricultural losses stemming from diseases reach a minimum of 10% annually [1], thus, early diagnosis of crop diseases and prevention are imperative to mitigate the spread of crop diseases [2]. The traditional diagnostic methods for the crop diseases heavily rely on visual assessments performed by agricultural workers or experts [3], which are highly subjective and inefficient. Consequently, the development of efficient and accurate disease recognition methods has become paramount.

**Data availability statement:** The datasets used are free and open. PlantVillage dataset (https://figshare.com/articles/dataset/PlantVillage_dataset/28234004?file=51771071). Potato Leaf Dataset (https://figshare.com/articles/dataset/potato_leaf_disease_rar/28233989?-file=51771044). The model code can be accessed at the following link: https://figshare.com/articles/software/dualbranch-/28267862.

**Funding:** The author(s) received no specific funding for this work.

**Competing interests:** The authors have declared that no competing interests exist.

In the leaves of crops infected by disease, the lesion areas often exhibit corresponding pathological features, which are the diagnostic criteria for crop diseases. In early studies, disease classification primarily relied on machine learning methods such as Support Vector Machine (SVM) and K-means clustering. For example, Zhang et al. [4] used Genetic Algorithm (GA) to optimize SVM and proposed GA-SVM to classify corn diseases. Their classifier achieved an accuracy rate of 92.82%. This work [5] employed the K-means clustering method to segment images of diseased parts and classified cucumber diseases using Sparse Representation (SR), and achieved an accuracy of 85.7%. Other traditional machine learning methods, such as K-Nearest Neighbors (KNN) [6], Random Subspace Method [7], Artificial Neural Networks (ANN) [8] and Random Forest (RF) [9] have also been widely used in various disease classification tasks and obtained impressive performance. The machine learning methods possess the advantages of strong interpretability and fast computational speed, but they require manual feature extraction, which limits their performance as the manually selected feature extraction algorithms may not be able to extract the optimal features[10]. Furthermore, designing feature extraction algorithms is time-consuming and labor-intensive, and different feature extraction algorithms may need to be selected for different diseases, resulting in poor universality of machine learning methods.

Convolutional Neural Networks [11] (CNNs) have dominated the field of computer vision. Composed primarily of convolutional layers and pooling layers, CNNs can automatically extract image information. Owing to their powerful representation capabilities, CNNs have become the preferred choice for agriculture. Sladojevic et al. [12] used a pre-trained CaffeNet to train on an Internet dataset containing 15 types of leaves, and achieved an average classification accuracy of 96.30%, which highlights the tremendous potential of CNNs. Chen et al. [13] integrated Inception blocks at the end of VGGNet, enabling the model to learn richer feature representations. Their model achieved a classification accuracy of 92.00% on a self-built rice dataset. Liang et al. [14] selected ResNet50 as the baseline and introduced Shuffle Units into the residual units, and obtained an accuracy of 98.00% on a mixed dataset. This work [15] constructed a lightweight LDSNet model by incorporating improved coordinated attention into densely dilated convolution blocks, and they reached a classification accuracy of 94.50% on a corn dataset. Ni et al. [16] introduced the Squeeze-and-Excitation (SE) module based on ResNet18 and modified the classifier structure, proposing a novel TomatoNet model that achieved a classification accuracy of 99.63% on a tomato dataset.These studies have all demonstrated the superior performance of CNNs in agricultural disease recognition.

The Transformer [17] is a standard paradigm within the domain of natural language processing. Dosovitskiy et al. [18] successfully applied the transformer architecture to the realm of computer vision and proposed the convolution-free model known as Vision Transformer (ViT). This model splits the input image into patches and feeds them into cascaded encoders to capture long-range semantic information. ViTs have demonstrated remarkable performance in downstream tasks like image classification and object detection. The remarkable performance exhibited by ViTs has spurred extensive research efforts aimed at their application in agricultural disease recognition. Thai et al. [19] attempted to use the standard ViT for cassava disease classification. They used Least Important Attention Pruning and sparse matrix multiplication to optimize the model's computation. The research results showed that this method outperformed existing advanced models at the time. Zhang et al. [20] employed the Swin Transformer architecture to classify a self-compiled rice disease dataset, and their model achieved an accuracy of up to 93.4%. In addition to directly using the standard models, some research efforts have personalized models based on task requirements. For instance, Yang et al. [21] designed a three-branch model based on Swin-Transformer, which can simultaneously predict disease types and severities. They utilized compact bilinear pooling techniques to achieve feature fusion across

different branches. On the AI Challenger 2018 dataset, this method achieved optimal performance. Chang et al. [22] designed an edge feature guidance module to enhance the extraction of edge information; they combined this module with Swin-Transformer and achieved a classification accuracy of 99.5%. Li et al. [23] proposed a dual-branch model based on the deformable attention Transformer that can consider both frequency domain information and global spatial domain information. Their model achieved the highest accuracy of 96.7% on the collected dataset. These works strongly demonstrate the effectiveness of ViTs in agricultural disease classification tasks, and also provide strong support for related research.

The lesion areas on crop leaves vary in size and often exhibit complex textures and shapes, posing challenges for high-precision recognition by deep learning models. However, both CNNs and ViTs have their own shortcomings. On the one hand, the receptive field of CNNs is limited by the size of the convolutional kernel, which makes it difficult to learn the global representation of images. On the other hand, ViTs do not perform well in learning local features such as textures and edges of diseases. Moreover, the high number of parameters and computational demands of ViTs make them unsuitable for resource-limited devices commonly used in real-world agricultural production. However, the deficiencies of ViTs are precisely the advantages of CNNs, and there have been efforts to combine the two to enhance feature extraction capabilities. In crop disease classification, four-stage hybrid models are commonly used, with hybrid methods mainly including serial hybrid and parallel hybrid [24]. For instance, Li et al. [25] proposed a serial model named ConvViT that combines CNN and ViT. In each stage of the model, the convolutional structure is first used to extract local features of the image, and then the Transformer structure is utilized to capture global features. Yua et al. [26] proposed ICVT, a model that integrates the Transformer architecture with the Inception convolutional framework. The first three stages of the model use deep convolutional transformer blocks, while the last stage employs Inception transformer blocks. Wang et al. [27] presented a four-stage parallel model named ShuFormer, which employs a parallel decoupled architecture. In each stage, intermediate features extracted by independent branches are fused through 1×1 convolutional blocks. These studies provide new perspectives for disease classification models. However, they either fail to fully consider the more complete fusion of local and global information or the number of parameters and complexity of the models still need to be optimized.

To address the aforementioned issues, this paper proposes a lightweight hybrid model that skillfully integrates the advantages of CNNs and ViTs. This model comprises two branches: CNN and Transformer. The former focuses on extracting local features, while the latter is responsible for capturing global features. By utilizing a adaptive parameter, a better fusion of local feature extraction and global context modeling is achieved, enabling the model to more effectively identify disease areas. In the Transformer block, an Aggregated Local Perception Feed-Forward Network (ALP-FFN) is introduced, which takes into account the extraction of spatial and channel information. A linear self-attention method is used, effectively reducing the memory and computational costs of the model. To validate the performance of the model, experiments were conducted on the PlantVillage dataset and the Potato Disease Leaf dataset.

This paper makes the following key contributions:

(1) We propose a dual-branch model for crop disease classification, which combines the advantages of CNNs and ViTs.

(2) We introduce ALP-FFN to introduce locality into the Transformer module.

(3) We construct a lightweight Transformer block using a linear self-attention and ALP-FFN, which reduces the model's memory consumption and computational cost while maintaining its superior performance.

(4) The proposed method achieves classification accuracies of 99.71% and 98.78% on two datasets, respectively, while reducing the number of parameters.

The following sections are organized as follows: Section 2 describes the datasets used in this paper and the proposed method for disease identification. Section 3 presents the experimental results and discussions. Finally, Section 4 summarizes the entire article and outlines future directions.

## 2. Materials and methods

### 2.1. Datasets and preprocessing

**2.1.1. Plantvillage dataset.** The PlantVillage dataset [28] is currently the largest open-source image classification dataset for plant diseases, which was carefully photographed by technical personnel in a laboratory setting and was professionally classified to promote the development of the agricultural computer vision. It consists of 54,305 images of 38 different types of leaves with dimension of 256 × 256 × 3.

**2.1.2. Potato Leaf Dataset.** The Potato Leaf Dataset [29] originates from Pakistan and consists of 4062 potato leaf images captured by different devices. It is necessary to point out that this dataset includes 1,628 images of early blight leaves, 1,414 images of late blight leaves, and 1020 images of healthy leaves with a resolution of 256 × 256 × 3.

Fig 1 depicts sample examples from the datasets.

In this paper, images were uniformly resized to 224×224 and normalized with mean and standard deviation. The two datasets were randomly split into training sets and test sets based on a ratio of 8:2.

### 2.2. Proposed model

The overall architecture of the proposed model is illustrated in Fig 2. It follows the hierarchical pyramid structure of the Swin Transformer [30], which consists of four stages, a global average pooling layer, and a fully connected layer. Each stage in the proposed model comprises a downsampling module and several sequentially stacked Global-Local Interactive Perception Unit (GLU) blocks processing data of the same shape. When an image is input into the first stage of the model, its height and width are reduced to one-quarter of their original sizes by the first downsampling module, and the number of the channels is expanded to the dimension C . After that, the feature map is input into the Transformer block with several layers. In the same way, the feature map is sequentially fed into the next three stages,which

PlantVillage Dataset 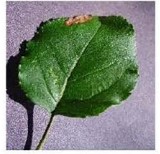 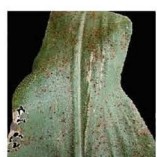 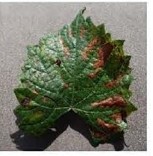 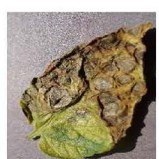 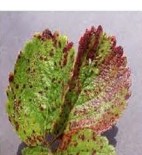 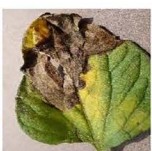

Potato Disease Dataset 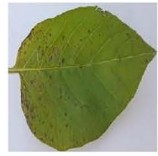 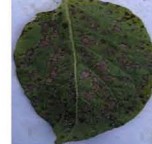 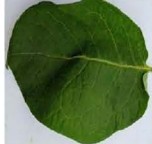 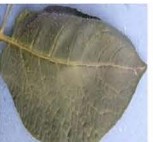 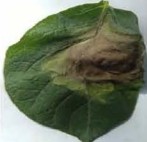 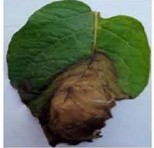

**Fig 1. Partial samples selected from the datasets.**

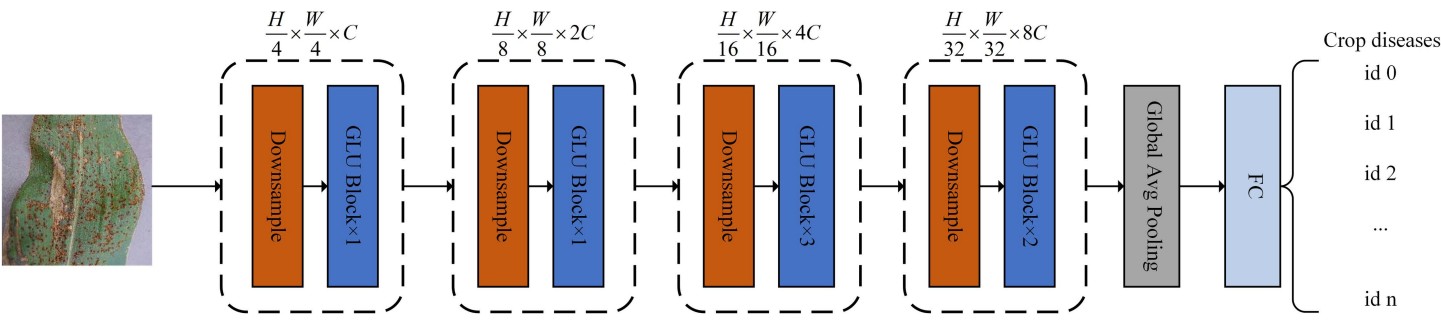

**Fig 2. Overall architecture of the proposed model.**

correspond to 8×, 16×, and 32× downsampling, respectively. Finally, the global average pooling layer and the fully connected layer are used to calculate the model's prediction confidence for each category. This multi-stage design not only generates multi-scale information similar to modern CNNs but also significantly reduces the computational cost associated with the self-attention mechanism.

## 2.3. GLU

The GLU is proposed and depicted in Fig 3. The input $X \in \mathbb{R}^{H \times W \times C}$ is fed into two parallel branches named as a convolutional block and a Transformer block to perform the parallel computation of the local and the global information, respectively. The output of the GLU is presented by the following formula:

$$X_{fused} = \lambda \times X_{transformer} + (1-\lambda) \times X_{cnn} \tag{1}$$

where $X_{fused}$ denotes the fusion result of the two branches, $X_{transformer}$ represents the computational output of the self-attention branch, and $X_{cnn}$ indicates the computational result of the

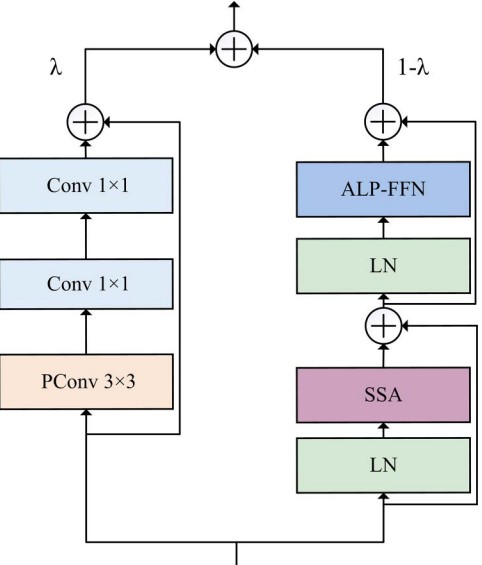

**Fig 3. Architecture of the GLU.**

convolutional branch. The learnable parameter $\lambda$ is employed to fuse the feature maps from the two branches. At the beginning of the training, $\lambda$ is initialized to 0.5, which indicates that the weights of these two branches are equal. As the training proceeds, $\lambda$ is dynamically adjusted through the backpropagation to allocate the larger weight to the more important branch. This module can flexibly adjust the fusion ratio of local and global information in each GLU block, thereby achieving better feature complementarity.

**2.3.1. Convolutional branch.** The self-attention mechanism of the Transformer architecture allows the model to capture the global information. However, its effectiveness in processing the local information is constrained. Therefore, a convolutional branch is constructed to introduce an inductive bias and to strengthen the model's local modeling ability. This branch is composed of $1 \times 1$ and $3 \times 3$ convolutions and illustrated in Fig 4.

The input image for the convolutional block is $X \in \mathbb{R}^{H \times W \times C}$, where H and W represent the height and the width of the input image, and C represents the number of the channels of the input image. When X enters the convolutional block, a $3 \times 3$ Partial Convolution (PConv) [31] operation is first performed to aggregate the local information of the input image. Subsequently, a point-wise convolution is used to fuse the channel information and to expand the number of the channels up to $4C$. Finally, another point-wise convolution is adopted to map the number of the channels back to C. A residual connection is established between the original input and the final output. To ensure the consistency of the image shape at each stage, the stride of the convolutional operations is set to 1 with pooling layers avoided.

PConv is used in the proposed model, which has fewer parameters and lower computational intensity. Given the redundancy in the channel information of the feature map, for the input image $X \in \mathbb{R}^{H \times W \times C}$, only the first quarter of the channels $X_{conv} \in \mathbb{R}^{H \times W \times \frac{C}{4}}$ are processed, the remaining channels $X_{identity} \in \mathbb{R}^{H \times W \times \frac{3C}{4}}$ are kept unchanged. Once $X_{conv}$ has completed the spatial information extraction, it will be concatenated with $X_{identity}$ for the subsequent computations.

$$Y_{conv} = PConv(X_{conv}) \tag{2}$$

$$Y = \left[ Y_{conv} ; X_{identity} \right] \tag{3}$$

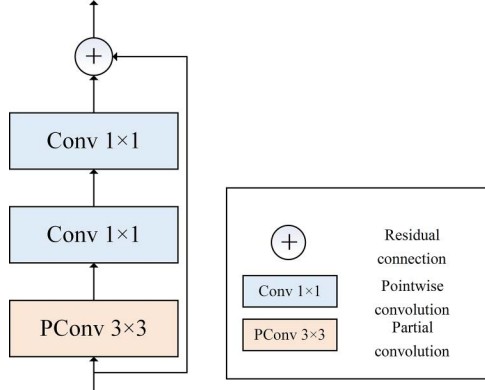

**Fig 4. Architecture of the convolutional branch.**

Assuming that the number of channels in the input and output feature maps is the same, the parametric quantities of the ordinary convolution and the PConv are represented by the following formulas.

$$Conv = K \times K \times C \times C \tag{4}$$

$$PConv = \frac{K \times K \times C \times C}{16} \tag{5}$$

where $K$ denotes the size of the convolution kernel, $C$ stands for the number of the input channels. These formulas demonstrate that the PConv employs fewer parameters than the ordinary convolution.

**2.3.2. Transformer branch.** The Transformer block is composed of self-attention layer and ALP-FFN layer and shown in Fig 5. The self-attention operation processes the spatial information, while the ALP-FFN operation fuses the channel information. Layer Normalization (LN) is applied before the self-attention layer and ALP-FFN layer, and residual connections are employed after these two components.

Initially, a 2D position embedding is added to the input feature map $X \in \mathbb{R}^{H \times W \times C}$ to retain the positional information. Then, the feature map is reshaped into a sequence of non-overlapping patches which serve as the input for the Transformer blocks.

$$Z_0 = X + E_{pos2d} \tag{6}$$

where $Z_0 \in \mathbb{R}^{N \times C}$ and $N = H \times W$.

Upon entering the Transformer block, the sequence undergoes the LN, followed by the self-attention computation to facilitate the interaction of the global information. The resultant output is then added to the input of the self-attention component. Following the self-attention computation, the channel information is integrated by the ALP-FFN layer.

$$z_l' = SA(LN(z_{l-1})) + z_{l-1}, l = 1,2 \ldots L \tag{7}$$

$$z_l = ALP - FFN(LN(z_l')) + z_l', l = 1,2 \ldots L \tag{8}$$

Self-attention is fundamental in the Transformer architecture. Given an input $Z \in \mathbb{R}^{N \times C}$, it is initially transformed into $q$, $k$, and $v$ matrices.

$$[q,k,v] = zW_{qkv} \tag{9}$$

where $W_{qkv} \in \mathbb{R}^{C \times 3C_k}$ is a learnable linear projection matrix.

The similarity between the matrices of $q$ and $k$ is computed through a dot product operation. The dot product is then divided by a scaling factor $\sqrt{C}$ to enhance the gradient stability during the training. The softmax function is applied for normalization to obtain the weight of the self-attention.

$$A = soft\max\left(\frac{qk^T}{\sqrt{C}}\right), A \in \mathbb{R}^{N \times N} \tag{10}$$

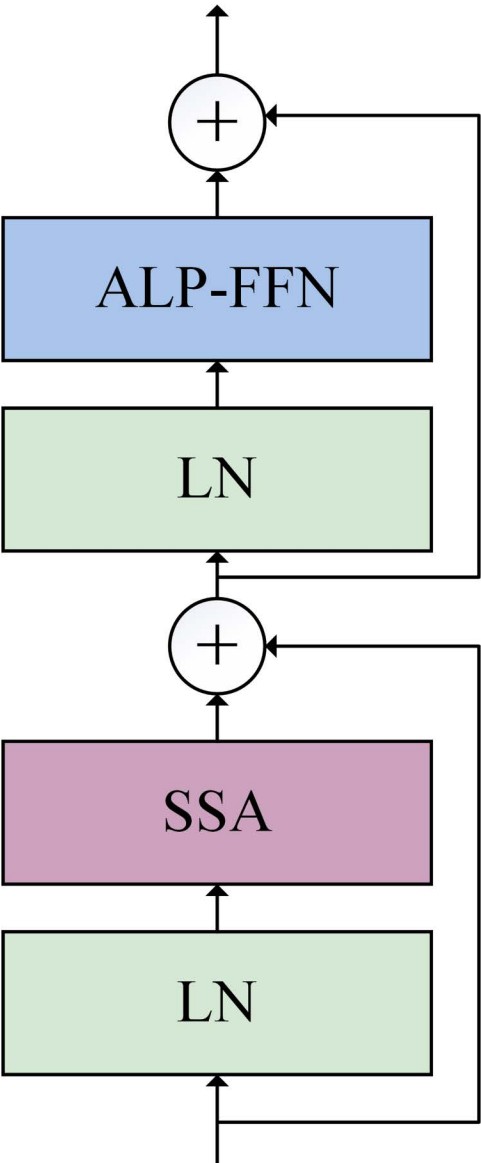

**Fig 5. Architecture of the transformer branch.**

The final output of the self-attention is the weighted sum of the attention matrix $A$ and the value matrix $v$.

$$SA(z) = Av \tag{11}$$

In practical applications, the multi-head self-attention mechanism is often adopted to learn different types of dependency relations. This mechanism employs $h$ attention heads to map the input z into different $q$, $k$, $v$ subspaces. The attention of $h$ heads is calculated in parallel, and the outputs of these heads are concatenated. Finally, the information is aggregated through a learnable linear projection.

$$MSA(z) = \left[ SA_1(z); SA_2(z); \cdots SA_h(z) \right] W^o \tag{12}$$

where $W^o \in \mathbb{R}^{C \times C}$.

The computational cost of the multi-head self-attention operation is enormous, which is not conducive to the training and inference of the model. Therefore, a Separable Self-Attention (SSA) [32] with linear complexity is introduced to to reduce the memory and computational cost, as shown in Fig 6.

Similarly, the input is mapped to three matrices, namely $q$, $k$, and $v$.

$$[q,k,v] = zW_{qkv}, \quad W_{qkv} \in \mathbb{R}^{C \times (2C+1)} \tag{13}$$

where $q \in \mathbb{R}^{N \times 1}, k, v \in \mathbb{R}^{N \times C}$

The softmax operation is applied to vector $q$ to obtain the context scores $c_s \in \mathbb{R}^k$, which incorporates the importance of each token.

$$c_s = soft\max(q) \tag{14}$$

The $c_s$ are then weighted and summed with matrix $k$ to obtain a context vector $c_v$, which encapsulates the importance of each channel.

$$c_v = \sum_{i=1}^{N} c_s(i)k(i) \tag{15}$$

Finally, the $c_v$ is multiplied by matrix $v$ using broadcast multiplication, and a linear layer is utilized to fuse channel information and produce the final output.

$$SSA = (c_v * v)W^o \tag{16}$$

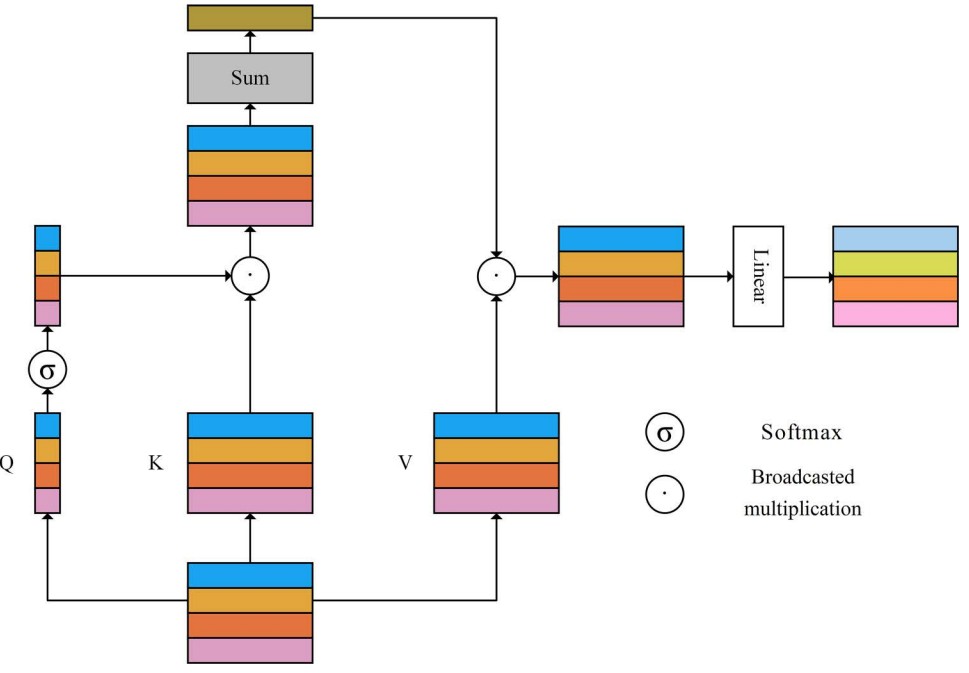

**Fig 6. Architecture of the SSA module.**

where, $W^o \in \mathbb{R}^{C \times C}$ and * represents broadcast multiplication.

Assuming an input of $X \in \mathbb{R}^{N \times C}$, for the traditional self-attention operation, strict matrix multiplication leads to a computational complexity of $O(N^2)$, where $N$ represents the length of the input sequence. Evidently, as the length of the input sequence increases, the computational cost rises sharply. However, in the SSA, the computational complexity is reduced to $O(N)$ due to the utilization of the broadcast multiplication instead of the rigorous matrix multiplication, and the computational complexity depending linearly on the length of the input sequence reduces the costs of training and inference. In addition, when performing linear projection to obtain q, SSA maps each vector of the input to a scalar instead of a vector, thereby reducing the number of parameters.

**2.3.3. ALP-FFN module.** The traditional FFN layer performs dimensionality reduction, dimensionality expansion, and nonlinear transformation, but it neglects the correlation between adjacent pixels. Inspired by the previous works[33,34], we propose the ALP-FFN module, which utilizes the convolutional operations instead of the fully connected layers to perform dimension mapping and nonlinear transformations. A detailed explanation is illustrated in Fig 7.

Given the tokens $x_t^h \in \mathbb{R}^{N \times C}$ at the h-th layer input, they are first restored to the "pictures" of $x_p^h \in \mathbb{R}^{\sqrt{N} \times \sqrt{N} \times C}$ based on their spatial dimensions representing the original image positions. For the transformed feature map, a point-wise convolution is first applied to expand number of channels to four times its original size, yielding $x_p^{l_1} \in \mathbb{R}^{\sqrt{N} \times \sqrt{N} \times 4C}$. Subsequently, a $3 \times 3$ DepthWise Convolution (DWConv) is used to enhance the interaction between the neighboring tokens, resulting in $x_p^d \in \mathbb{R}^{\sqrt{N} \times \sqrt{N} \times 4C}$. Finally, a point-wise convolution is applied to restore the number of channels, producing $x_p^{l_2} \in \mathbb{R}^{\sqrt{N} \times \sqrt{N} \times C}$. Each convolution operation is followed by Batch Normalization (BN) and Gaussian Error Linear Unit (GELU) activation function. Once the information extraction is complete, the feature map is flattened back into the tokens: $x_p^f \in \mathbb{R}^{N \times C}$. The Context Broadcasting (CB) module [35] is inserted at the end of the feed-forward network, and its calculation formula is as follows:

$$CB(x_i) = \frac{x_i + \frac{1}{N}\sum_{j=1}^{N} x_j}{2} \qquad (17)$$

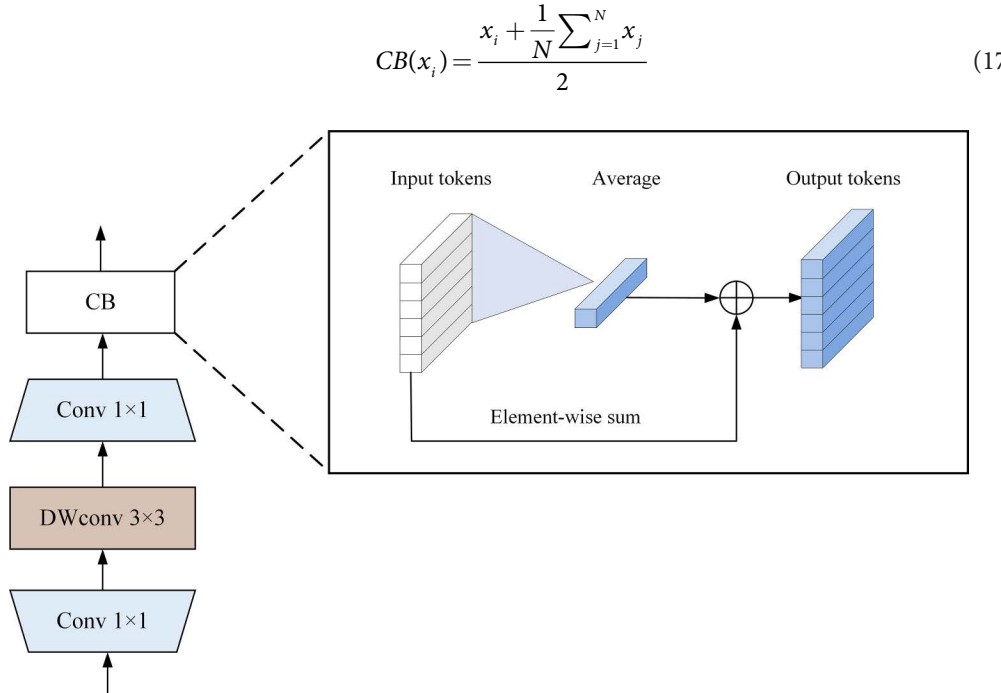

**Fig 7. Architecture of the ALP-FFN module.**

where $x_i$ represents the $i^{th}$ token. By taking the average of all tokens, an aggregated token is obtained. Subsequently, this token is broadcast-added to the original tokens, propagating the aggregated information back to the original tokens. This module can inject uniform attention into the model, thus alleviating the problem of the model's difficulty in learning dense attention maps.

In summary, the computational process of the ALP-FFN module can be represented as follows:

$$x_p^h = \text{SpatialRestore}(x_t^h) \tag{18}$$

$$x_p^{l_1} = \text{GELU}(\text{BN}(\text{Linear1}(x_p^h))) \tag{19}$$

$$x_p^d = \text{GELU}(\text{BN}(\text{DWConv}(x_p^{l_1}))) \tag{20}$$

$$x_p^{l_2} = \text{GELU}(\text{BN}(\text{Linear2}(x_p^d))) \tag{21}$$

$$x_p^f == \text{Flatten}(x_p^{l_2}) \tag{22}$$

$$x_t^{h+1} = \text{CB}(x_p^f) \tag{23}$$

unlike traditional FFN layers, ALP-FFN introduces depth-wise convolutional operations to extract spatial local information, enhancing the learning ability for local features. Meanwhile, the CB mechanism can propagate global information to each token, improving the model's capacity and generalization.

## 3. Results and discussion

### 3.1. Experimental settings

All experiments were conducted on Ubuntu 20.04, using an Intel(R) Xeon(R) Platinum 8352V CPU @ 2.10GHz. Training was accelerated with an NVIDIA GeForce RTX 4090 graphics card with 24GB of video memory. CUDA version 11.3 was utilized, and the proposed model was implemented using Pytorch 1.10.0.

In the experiments, the hyperparameters were primarily set based on the DeiT [36] approach. The batch size was set to 256, and the model was trained from scratch for 100 epochs. During the first five epochs, the learning rate warmed linearly from 0.000001 up to 0.01, followed by a cosine decay scheduler. The AdamW optimizer was used with a weight decay of 0.05. To enhance the model's generalization ability, rigorous data augmentation schemes were applied to the training images, including automatic augmentation, random erasing, Mixup, and CutMix techniques. Additionally, label smoothing and stochastic depth were employed to prevent overfitting.

### 3.2. Performance metrics

Model performance is evaluated using Accuracy, Recall, Precision, F1-score, the number of parameters and Floating Point Operations (FLOPs). Accuracy is used to assess the overall

performance of the model. Precision represents the credibility of the model's prediction for a particular disease. Recall demonstrates how many diseased leaves can the model predict. F1-score is the harmonic mean of Precision and Recall, which is used to strike a balance between these two metrics. The number of parameters is used to evaluate the complexity of the model, the larger the number, the more complex the model. FLOPs is used to measure the computational cost of the model, which represents the number of floating point operations required for the model to perform a forward propagation. The higher the FLOPs, the longer the computation time required for the model. The calculation formulas are specified as follows:

$$Accuracy = \frac{TP + TN}{TP + TN + FP + FN} \tag{24}$$

$$Recall = \frac{TP}{TP + FN} \tag{25}$$

$$Precision = \frac{TP}{TP + FP} \tag{26}$$

$$F1 - score = \frac{2 \times Precision \times Recall}{Precision + Recall} \tag{27}$$

where, True Positive (TP) and True Negative (TN) refer to the number of correctly predicted positive and negative samples, respectively. False Positive (FP) represents the number of negative samples mistakenly identified as positive by the model, while False Negative (FN) represents the number of positive samples wrongly predicted as negative.

We compared our proposed model with the well accepted eight state-of-the-art visual models. Among them, ResNet [37], RepVGG [38] and MixNet [39] were selected and stood for the CNN-based models. ViT, LeViT [40], MobileViT-v2 [32], PVTv2 [41] and TNT [42] were chosen and represented the ViT-based models. The selected models were loaded through the timm library, and the number of parameters and FLOPs were calculated through the THOP library.

**3.2.1. Experimental results on the PlantVillage dataset.** The experimental results on the PlantVillage dataset for the mainstream CNNs and ViTs are listed in Table 1. The proposed model achieved the highest classification accuracy of 99.71%, and its Recall, Precision and F1-score reached 99.52%, 99.45%, and 99.48%, respectively. These four metrics outperform those of the other eight models. Compared to the second best performing model TNT-S, our model improves accuracy by 0.6%, Recall, Precision and F1-score by 0.79%,0.80% and 0.80% respectively. These excellent performances indicate that the model well complements the local and global features; the ALP-FFN module enables the Transformer branch to model locally as well, which allows the model to extract richer information. Additionally, the model also excels in terms of lightweight performance. Compared with TNT-S which owns the second-best performance, the proposed model achieves better performance with only 21.11% parameters and 12.78% FLOPs. The reduced parameters and FLOPs originate from the introduced PConv and SSA module.

Table 2 presents a comparison between our model and previous works. It can be seen that our method has reached an advanced level, which further verifies the effectiveness of the proposed model. Chen et al. constructed a novel network, Es-MbNet, which integrates

**Table 1. Comparison of different networks on the PlantVillage dataset.**

| Models | Accuracy (%) | Recall (%) | Precision (%) | F1-score (%) | Params (M) | FLOPs (G) |
|---|---|---|---|---|---|---|
| Resnet-18 | 98.05 | 98.57 | 98.52 | 98.53 | 11.20 | 1.82 |
| MixNet-L | 98.29 | 97.50 | 97.74 | 97.58 | 5.80 | 0.55 |
| RepVGG-A2 | 96.95 | 95.55 | 96.42 | 95.88 | 26.81 | 5.68 |
| ViT-S | 96.42 | 96.89 | 96.52 | 96.70 | 21.38 | 16.76 |
| LeViT-256 | 98.87 | 98.59 | 98.69 | 98.60 | 17.90 | 1.06 |
| MobileViTv2 | 97.11 | 95.90 | 96.78 | 96.14 | 9.82 | 3.15 |
| PVTv2-B1 | 96.04 | 94.82 | 95.97 | 95.16 | 13.52 | 2.04 |
| TNT-S | 99.11 | 98.73 | 98.65 | 98.68 | 23.31 | 4.85 |
| Proposed | 99.71 | 99.52 | 99.45 | 99.48 | 4.92 | 0.62 |

**Table 2. Comparison of the proposed model with previous works.**

| Cases | Accuracy (%) | Recall (%) | Precision (%) | F1-score (%) |
|---|---|---|---|---|
| Singh et al. [43] | 98.86 | 98.90 | 98.81 | 98.85 |
| Li et al. [25] | 99.41 | 98.54 | 97.80 | 98.07 |
| Nandhini et al. [44] | 98.81 | 98.75 | 98.60 | 98.67 |
| Chen et al. [45] | 99.61 | 98.08 | – | 98.08 |
| Sahu et al. [46] | 98.90 | 93.60 | – | 97.80 |
| Kaya et al. [47] | 98.17 | 98.17 | 98.16 | 98.12 |
| Zhao et al. [48] | 99.00 | 98.60 | 98.50 | 98.40 |
| Proposed Model | 99.71 | 99.52 | 99.45 | 99.48 |

SE-MobileNet, Mobile-DANet, and MobileNet V2. This model achieved an average accuracy of 99.61% on the PlantVillage dataset. Our model, however, reached a classification accuracy of 99.71%, representing an improvement of 0.10% over theirs. Singh et al. proposed a serial model combining CNN and Transformer, consisting of two VGG16 blocks, one Inception v7 block, and four Transformer encoder blocks. The proposed model achieved a Recall of 98.90%, a Precision of 98.81%, and an F1-score of 98.85% on the dataset. In contrast, our proposed multi-stage parallel model achieved Recall, Precision, and F1-score of 99.52%, 99.45%, and 99.48%, respectively, representing improvements of 0.62%, 0.64%, and 0.63% over theirs.

**3.2.2. Experimental results on the Potato Leaf Dataset.** The experimental results of the proposed model on the Potato Leaf Dataset are listed in Table 3. The proposed model still achieved competitive performance. The recognition accuracy, recall, precision, and F1-score are 98.78%, 98.75%, 98.68%, and 98.72%, respectively. Compared to the ResNet-18 model, which has the highest accuracy, our model improves the classification accuracy by 0.73%. Additionally, the ResNet-18 model requires 23.31M parameters and 4.85G FLOPs. In contrast, our model only needs 4.9M parameters and 0.62G FLOPs, indicating a significant reduction in both parametric complexity and computational cost while maintaining higher accuracy.

## 3.3. Ablation experiments

The proposed model compared with the other advanced methods exhibits the superior performance. To delve further into the specific contributions from each module to the model's

**Table 3. Comparison of different networks on the Potato Leaf Dataset.**

| Models | Accuracy (%) | Recall (%) | Precision (%) | F1-score (%) | Params (M) | FLOPs (G) |
|---|---|---|---|---|---|---|
| Resnet-18 | 98.05 | 98.57 | 98.52 | 98.53 | 11.18 | 1.82 |
| MixNet-L | 87.82 | 86.93 | 88.34 | 87.45 | 5.74 | 0.55 |
| RepVGG-A2 | 93.05 | 93.21 | 92.98 | 92.90 | 26.76 | 5.68 |
| ViT-S | 62.00 | 57.55 | 70.39 | 57.72 | 21.37 | 16.76 |
| LeViT-256 | 91.35 | 89.31 | 90.65 | 89.83 | 17.86 | 1.06 |
| MobileVit-v2 | 97.80 | 99.62 | 99.76 | 97.69 | 9.79 | 3.15 |
| PVTv2-B1 | 96.46 | 96.53 | 96.26 | 96.37 | 13.50 | 2.04 |
| TNT-S | 95.73 | 95.22 | 96.07 | 95.59 | 23.30 | 4.85 |
| Proposed | 98.78 | 98.75 | 98.68 | 98.72 | 4.91 | 0.62 |

performance, a series of ablation experiments were designed to reveal the effectiveness of different modules in the model by deleting or replacing them.

**3.3.1. Effectiveness of the dual-branch architecture.** In this section, experimental comparisons between the CNN branch and the Transformer branch were conducted, and the results are listed in Table 4. The network that only contains the CNN branch achieved classification accuracies of 96.32% and 95.00% on the two datasets, while the pure Transformer network also achieved accuracies of 97.02% and 96.71%. The proposed method achieved better performance on both two datasets, which indicates that our approach can effectively extract both local and long-range dependencies.

Additionally, we investigate the fusion strategy of the Transformer and CNN branches in the GLU, involving the direct addition and the channel dimension concatenation, followed by the point-wise convolution mapping to the original dimension. The experimental results are listed in Table 5. The fusion method which adopts the direct addition decreases the classification accuracy by 1.33% and 1.78%, respectively. Because the direct addition fusion method considers both branches equally important and neglects their unequal actual contributions. The fusion method which adopts the channel dimension concatenation decreases the model performance by 1.73% and 0.98%, respectively. Here, too much channel redundant information is introduced. Furthermore, to map the dimensions, this approach introduces additional point-wise convolution kernels, resulting in an increase in the number of parameters. In contrast, the weighted strategy proposed in this paper can better achieve the fusion of local and global features, thereby achieving superior performance.

**3.3.2. Effectiveness of the ALP-FFN module and the SSA module.** To validate the effectiveness of the ALP-FFN and SSA modules, we replaced these two modules with the original MLP layer and self-attention layer respectively to observe changes in performance. The results are shown in Table 6, where AF means the model using ALP-FFN and SSA

**Table 4. Results of ablation experiments for the two branches of the model.**

| Dataset | Experiments | Accuracy (%) | Recall (%) | Precision (%) | F1-score (%) | Params (M) |
|---|---|---|---|---|---|---|
| PlantVillage | Proposed | 99.71 | 99.52 | 99.45 | 99.48 | 4.9 |
| | CNN | 96.32 | 94.85 | 95.31 | 94.95 | 2.2 |
| | Transformer | 97.02 | 95.53 | 96.21 | 95.77 | 2.9 |
| Potato Leaf | Proposed | 98.78 | 98.75 | 98.68 | 98.72 | 4.9 |
| | CNN | 95.00 | 94.83 | 95.05 | 94.93 | 2.2 |
| | Transformer | 96.71 | 96.85 | 96.41 | 96.61 | 2.9 |

**Table 5. Comparison of different fusion methods for the two branches of the model.**

| Dataset | Experiments | Accuracy (%) | Recall (%) | Precision (%) | F1-score (%) | Params (M) |
|---|---|---|---|---|---|---|
| PlantVillage | Add | 98.38 | 97.85 | 97.87 | 97.80 | 4.9 |
| | Concat | 97.98 | 97.22 | 97.32 | 97.19 | 5.3 |
| | Proposed | 99.71 | 99.52 | 99.45 | 99.48 | 4.9 |
| Potato Leaf | Add | 97.80 | 97.71 | 97.74 | 97.73 | 4.9 |
| | Concat | 97.80 | 97.80 | 97.75 | 97.77 | 5.3 |
| | Proposed | 98.78 | 98.75 | 98.68 | 98.72 | 4.9 |

**Table 6. Comparative experiments on different network model structures.**

| Dataset | AF | SSA | Accuracy (%) | Recall (%) | Precision (%) | F1-score (%) | Params (M) | FLOPs (G) |
|---|---|---|---|---|---|---|---|---|
| PlantVillage | | | 90.98 | 89.48 | 92.61 | 89.35 | 5.09 | 0.63 |
| | ✓ | | 94.22 | 91.76 | 94.42 | 92.48 | 5.15 | 0.65 |
| | | ✓ | 97.30 | 96.38 | 96.52 | 96.30 | 4.86 | 0.60 |
| | ✓ | ✓ | 99.71 | 99.52 | 99.45 | 99.48 | 4.92 | 0.62 |
| Potato Leaf | | | 95.74 | 95.93 | 95.43 | 95.65 | 5.08 | 0.63 |
| | ✓ | | 97.69 | 97.79 | 97.61 | 97.70 | 5.14 | 0.65 |
| | | ✓ | 96.71 | 96.97 | 96.44 | 96.67 | 5.85 | 0.60 |
| | ✓ | ✓ | 98.78 | 98.75 | 98.68 | 98.72 | 4.91 | 0.62 |

means the model using linear self-attention. It can be seen that the model using the original MLP layer and the self-attention layer performs the worst. On the PlantVillage dataset, the model's Accuracy, Recall, Precision, and F1-score were 90.98%, 89.48%, 92.61% and 89.35% respectively, while on the Potato Leaf dataset, these metrics were 95.74%, 95.93%, 95.43%, and 95.65% respectively. On both datasets, the ALP-FFN and SSA modules brought about varying degrees of performance improvement. The ALP-FFN module introduced locality into the Transformer branch, enabling the model to learn richer features. Models incorporating ALP-FFN achieved accuracy improvements of 3.24% and 1.95% on the PlantVillage and Potato Leaf datasets, respectively, with improvements also observed in recall, precision, and F1 score. Due to the introduction of DWConv, models using the ALP-FFN module had slightly increased parameter and computation counts compared to the original model. Models using SSA achieved accuracy improvements of 6.32% and 0.97% on the two datasets, respectively. Additionally, the SSA module employed scalar projection and element-wise operations, reducing parameter and computation counts by 0.23M and 0.03G, respectively. Notably, due to the complex batch matrix multiplication in the original self-attention, replacing linear self-attention with the original self-attention resulted in a sharp increase in the model's memory requirements, with the batch size during training and inference having to be adjusted from 256 to 8. In contrast, models integrating SSA could use larger batch sizes, significantly reducing training difficulty. When integrating the ALP-FFN and SSA modules, the model achieved maximum values across all performance metrics. Accuracy improvements of 8.73% and 3.04% were observed on the PlantVillage and Potato Leaf datasets, respectively. These results indicate that integrating both modules enhanced performance while reducing parameter count and complexity.

## PlantVillage Dataset          Potato Leaf Dataset

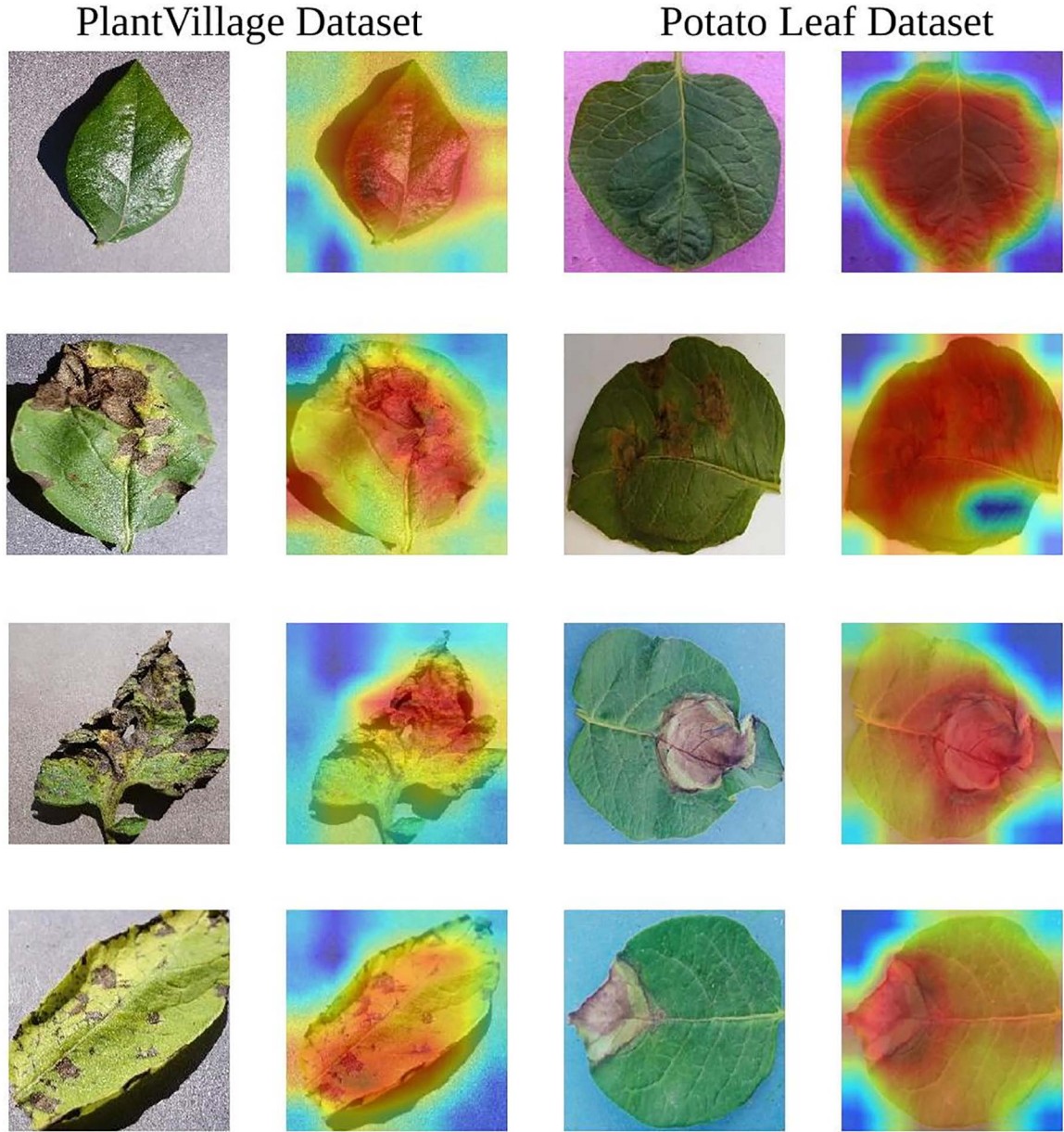

**Fig 8. Grad-CAM visualization results of the proposed model on the two datasets.**

### 3.4. Visualization analysis

**3.4.1. Grad-CAM analysis.** The Gradient-weighted Class Activation Mapping (Grad-CAM) [49] was used to visualize the features of the last normalized layer in the model, as shown in Fig 8. Here, the deeper color of the red region indicates the more attention paid by the model, while the yellow and the blue regions represent the less attention paid by the model. Comparing the original disease images with the heatmaps, we find that the red regions align closely with the lesion areas. For healthy leaves, the model focuses on the entire leaves. These outcomes collectively indicate the effectiveness of the proposed model.

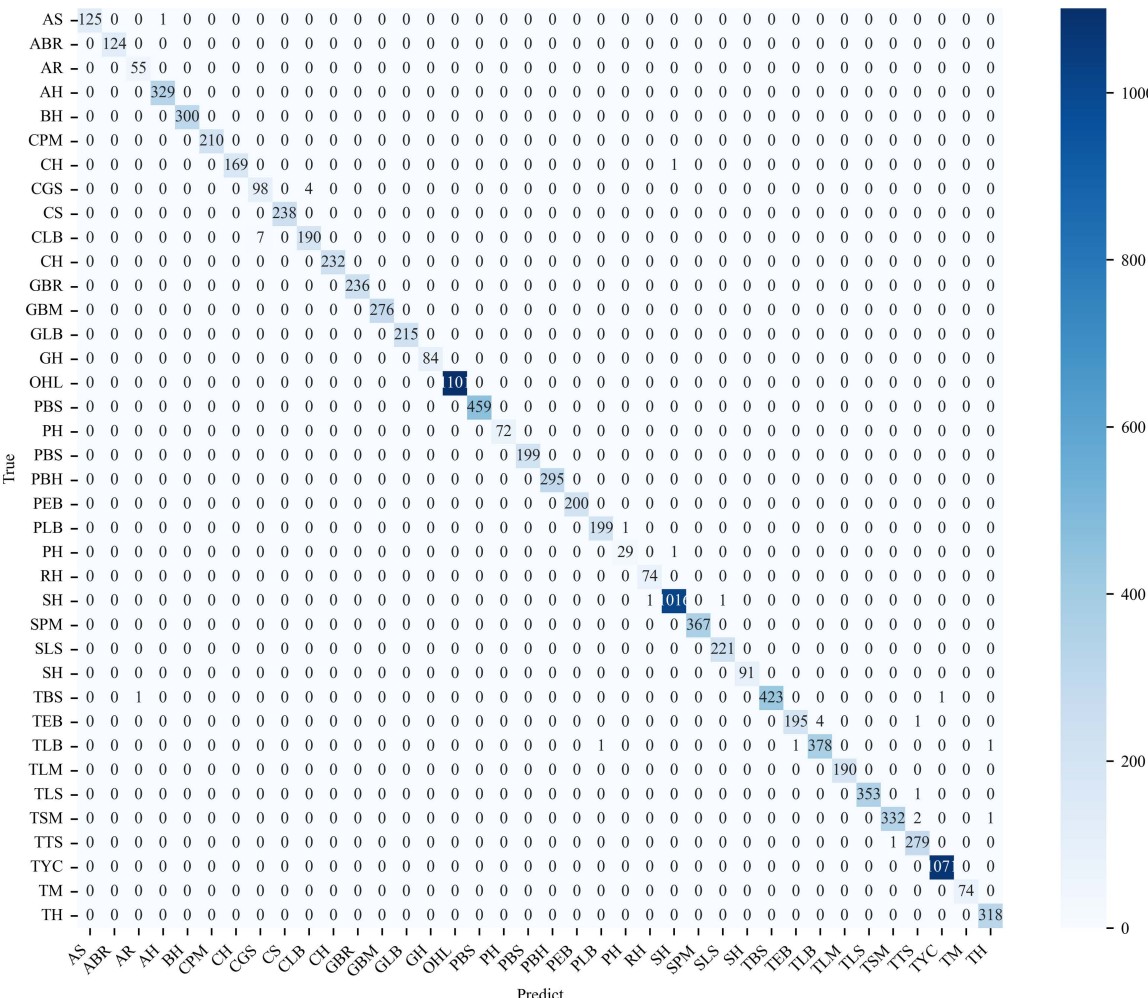

**Fig 9. Confusion matrix of the proposed model on the PlantVillage Dataset.**

**3.4.2. Confusion matrix.** This section presents the classification confusion matrix of the proposed model on the two datasets, as shown in Figs 9 and 10. In the confusion matrix, the Y-axis represents the actual image label, while the X-axis denotes the model's predicted label. The diagonal elements of the matrix indicate the instances classified correctly by the proposed model, the rest elements represent the misclassifications. The darker colors in the diagonal grid indicate the better classification results. Apparently, the majority of the samples are correctly predicted, and only a few minority is misclassified. The obtained classification confusion matrix of the proposed model indicates that the proposed model has not learned any significant biases.

**3.4.3. ROC analysis.** We employed the Receiver Operating Characteristic (ROC) to visualize the classification performance of the proposed model for each category. Due to the vast number of categories existed in the PlantVillage dataset, this section solely presents the visualization results for the Pota-to Leaf Dataset and shown in Fig 11. By iterating the True Positive Rate (TPR) and the False Positive Rate (FPR) under various threshold conditions, we generated a series of discrete points and plotted them in Fig 11. The horizontal axis of

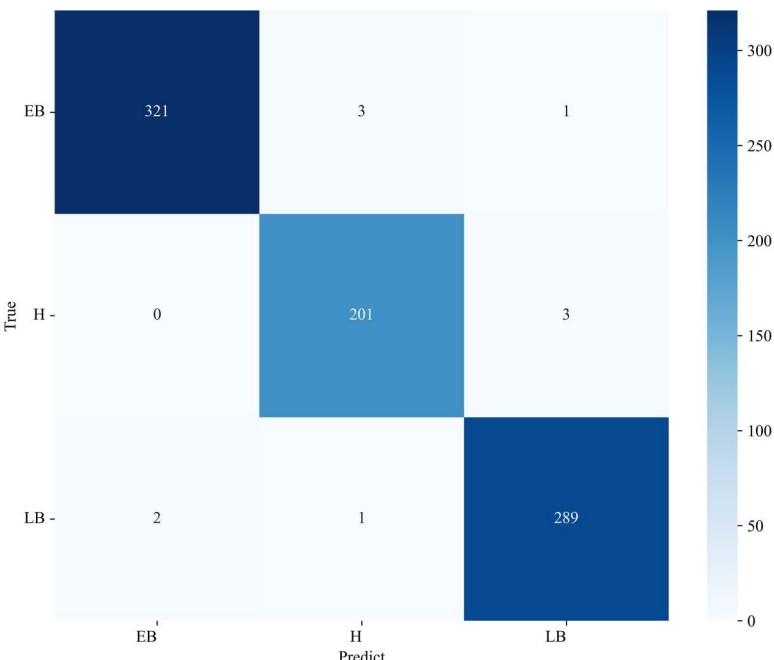

**Fig 10. Confusion matrix of the proposed model on the Potato Leaf Dataset.**

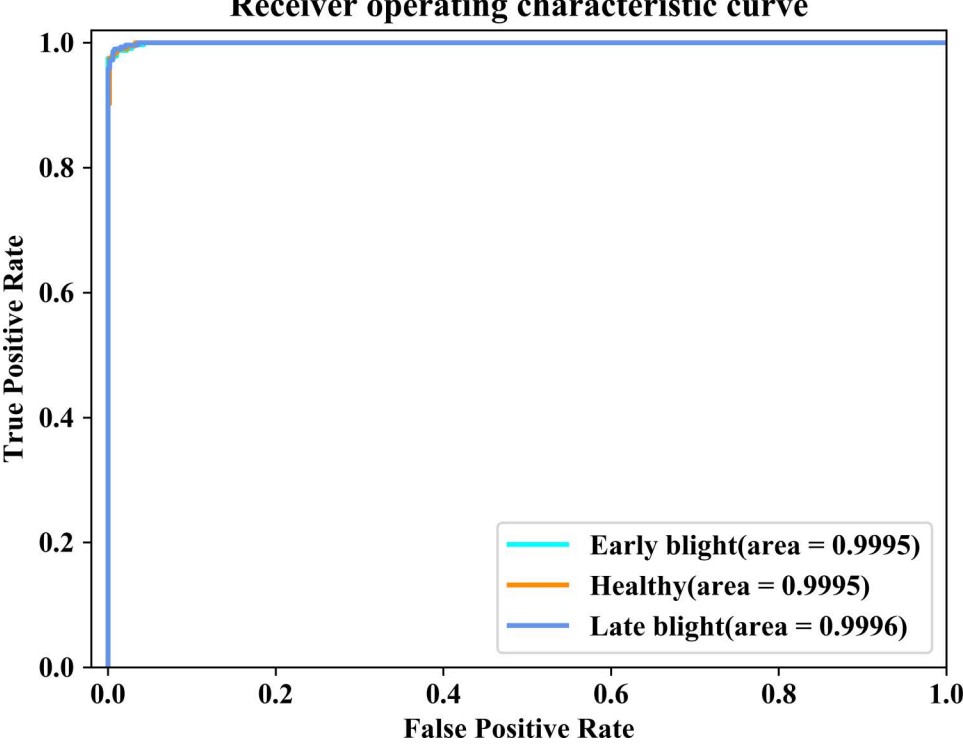

**Fig 11. ROC curves of the proposed model on the Potato Leaf Dataset.**

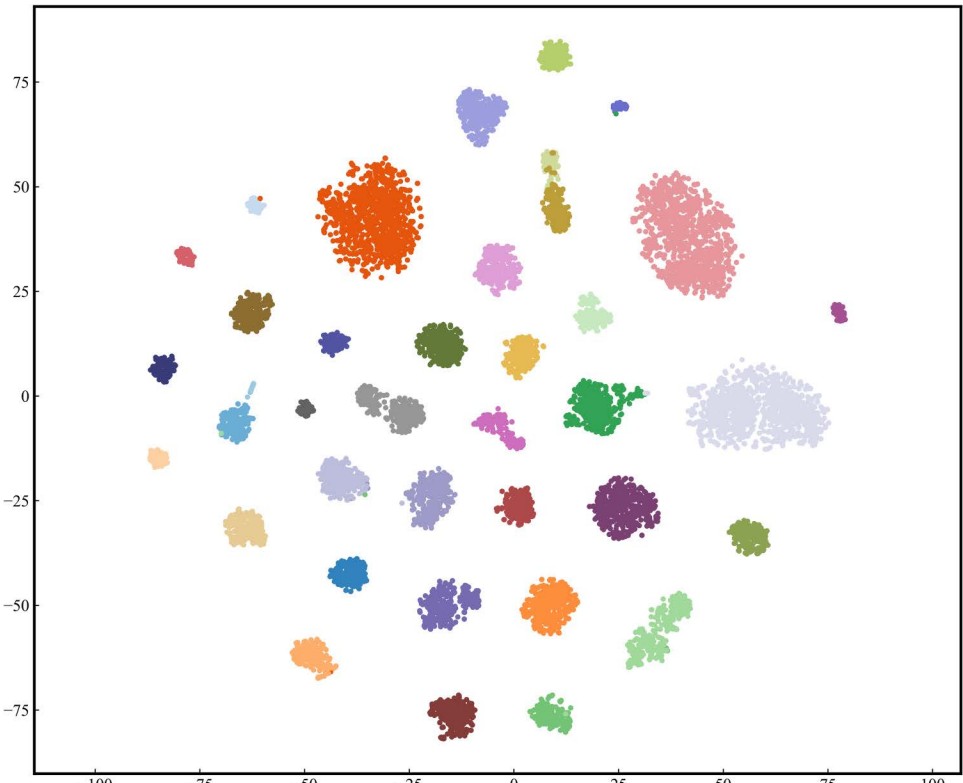

**Fig 12. T-SNE visualization results of the proposed model on the PlantVillage Dataset.**

Fig 11 represents the FPR and the vertical axis represents the TPR. Area Under Curve (AUC) is defined as the area enclosed by the ROC curve and the horizontal axis ranging from 0 to 1. A higher AUC value indicates the proposed model has a better performance. Apparently, the AUC values generated from all three categories of images are close to 1, which verified the effectiveness of the proposed model.

**3.4.4. t-SNE analysis.** The t-Distributed Stochastic Neighbor Embedding (t-SNE) [50] method was employed to visualize the relationships among the data. This algorithm maps the high-dimensional data learned by the proposed model into a two-dimensional space and preserves the feature distribution of the data in the high-dimensional space. Figs 12 and 13 demonstrate the visualization results of the proposed model on two datasets, in which each point represents a sample and each color rep-resents a type of disease. It is observed that the samples belong to different categories are clearly separated, while the samples belong to the same category are closely arranged. For the Potato Leaf Dataset, the model can clearly identify and distinguish the three types of diseases. Even in the PlantVillage dataset with more diverse disease types, the proposed model also demonstrates a good discriminative ability. These visualization results strongly validate the effectiveness of our proposed model in extracting different feature information.

## 4. Conclusions

This paper proposes a dual-branch model that integrates convolutional neural network and Transformer for precise identification of crop diseases. The convolutional branch focuses on extracting local features from images, while the Transformer branch is responsible for

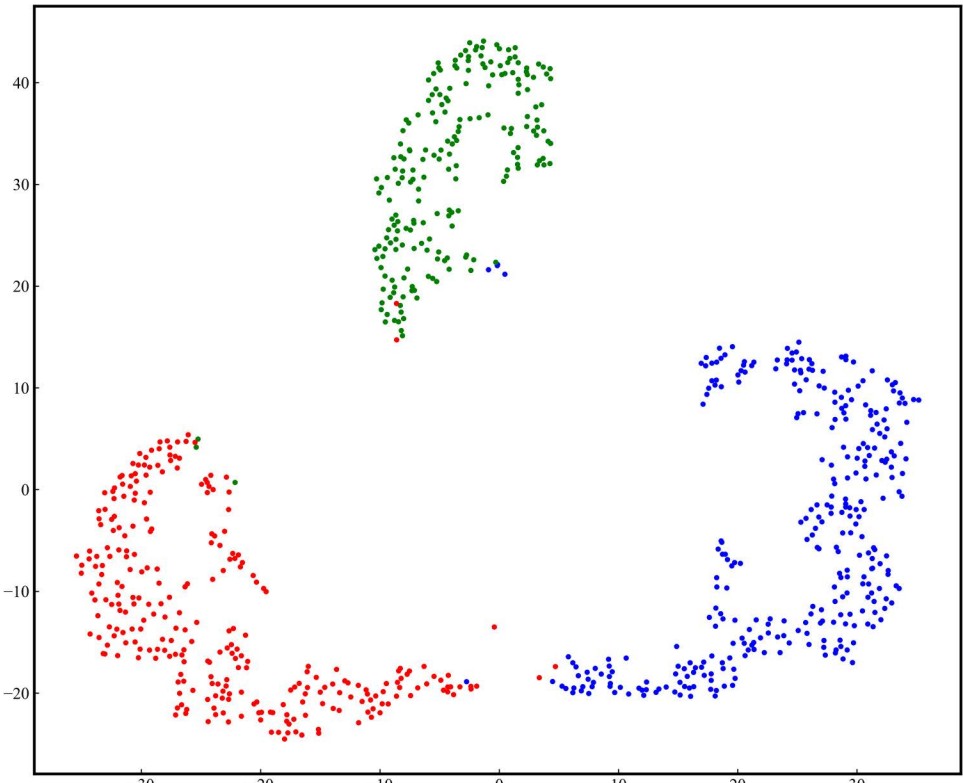

**Fig 13. T-SNE visualization results of the proposed model on the Potato Leaf Dataset.**

capturing global feature representations. The fusion of the two branches is dynamically controlled by a learnable parameter. This paper also introduces the ALP-FFN module, which injects local modeling capability into the Transformer branch for better model performance. Furthermore, this paper integrates the ALP-FFN module with a linear self-attention mechanism to propose a computationally efficient Transformer block. On two datasets, the model achieves classification accuracies of 99.71% and 98.78%, respectively, outperforming eight advanced models. These results indicate that the model can accurately identify various crop diseases, which is crucial for early disease intervention. Furthermore, the model has a parameter count of only 4.9M and a computational complexity of 0.62G, making it friendly to various resource-constrained devices. Future works are as follows: (1) Collecting dataset of diseases in real-world field environments to more thoroughly evaluate the performance of the model; (2) Extending the model to more crop species and disease types.

## Author contributions

**Conceptualization:** Qingduan Meng, Jiadong Guo, Hui Zhang.

**Formal analysis:** Jiadong Guo, Yaoqi Zhou.

**Resources:** Xiaoling Zhang.

**Software:** Xiaoling Zhang.

**Validation:** Hui Zhang.

**Visualization:** Yaoqi Zhou.

**Writing – original draft:** Jiadong Guo.

**Writing – review & editing:** Qingduan Meng.

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
