## [Decision Letter · Decision Letter 0]

8 Dec 2024

PONE-D-24-46420A dual-branch model combining convolution and Vision Transformer for crop disease classificationPLOS ONE

Dear Dr. Zhang,

Thank you for submitting your manuscript to PLOS ONE. After careful consideration, we feel that it has merit but does not fully meet PLOS ONE’s publication criteria as it currently stands. Therefore, we invite you to submit a revised version of the manuscript that addresses the points raised during the review process.

Also address the following comments while revising your paper:

Briefly mention how the model outperforms state-of-the-art approaches, emphasizing its advantages in accuracy and computational efficiency.Clarify the technical importance of integrating the ALP-FFN module and self-attention mechanism for improved performance.Provide context for the reported accuracy and efficiency, explaining how these advancements address specific challenges in crop disease identification or precision agriculture.

We look forward to receiving your revised manuscript.

Kind regards,

Venkata Krishna Parimala

Academic Editor

PLOS ONE

Journal Requirements:

3. Please ensure that you refer to Figure 1 in your text as, if accepted, production will need this reference to link the reader to the figure.

Reviewers' comments:

Reviewer's Responses to Questions

**Comments to the Author**

1. Is the manuscript technically sound, and do the data support the conclusions?

Reviewer #1: Yes

Reviewer #2: Yes

2. Has the statistical analysis been performed appropriately and rigorously? 

Reviewer #1: Yes

Reviewer #2: Yes

3. Have the authors made all data underlying the findings in their manuscript fully available?

Reviewer #1: Yes

Reviewer #2: Yes

4. Is the manuscript presented in an intelligible fashion and written in standard English?

Reviewer #1: Yes

Reviewer #2: Yes

5. Review Comments to the Author

Reviewer #1: 1- The most important pre-processing techniques for agricultural images are not shown.

2- Add additional research to review the previous research review for the research field

3- Please explain figure 7 better.

4- Discuss the results listed in the tables and the criteria that were adopted to establish a clear vision.

5- Clarification of proposed future work from this manuscript

Reviewer #2: 1. Your figures appears as an appendix, fix them within the papers even if it takes a whole page.

2. Your tables and format is not well arranged, make your tables in a page, not in separate pages.

3. Follow the actual format.

6. PLOS authors have the option to publish the peer review history of their article (what does this mean? ). If published, this will include your full peer review and any attached files.

**Do you want your identity to be public for this peer review?** For information about this choice, including consent withdrawal, please see our Privacy Policy .

Reviewer #1: No

Reviewer #2: No

---

## [Author Response · Author response to Decision Letter 1]

26 Jan 2025

Dear editor and reviewers:

Thank you for offering us an opportunity to improve the quality of our submitted manuscript “A dual-branch model combining convolution and Vision Transformer for crop disease classification”. We appreciate very much the editor's and reviewers’ constructive and insightful comments. In this revision, we have addressed all of these comments. We hope the revised manuscript has now met the publication standard of your journal.

Revised portion are marked in red in the paper. The main corrections in the paper and the responses to the editor's and reviewers’ comments are as following:

Editor

Thank you for submitting your manuscript to PLOS ONE. After careful consideration, we feel that it has merit but does not fully meet PLOS ONE’s publication criteria as it currently stands. Therefore, we invite you to submit a revised version of the manuscript that addresses the points raised during the review process. Also address the following comments while revising your paper:

1. Comment: Briefly mention how the model outperforms state-of-the-art approaches, emphasizing its advantages in accuracy and computational efficiency.

Reply: Thank you for your advice. We have highlighted the advantages of model performance as well as computational efficiency in the abstract, results discussion section, and conclusion.

2. Comment: Clarify the technical importance of integrating the ALP-FFN module and self-attention mechanism for improved performance.

Reply: We have supplemented the description of the advantages of linear self-attention and ALP‑FFN modules in section 2.3.2 and section 2.3.3, and conducted additional ablation experiments to verify the effectiveness of the two modules. As a result, the original section 3.3 is now divided into two parts: section 3.3.1 “effectiveness of the dual-branch architecture” and section 3.3.2 “effectiveness of the ALP-FFN module and the SSA module respectively”. Furthermore, we have removed the description of the ALP-FFN module’s parameter reduction from both the introduction and section 2.3.3, since newly added ablation experiments indicate that the deep convolution operation introduced by ALP-FFN slightly increases the number of parameters. We apologize for this oversight.

3. Comment: Provide context for the reported accuracy and efficiency, explaining how these advancements address specific challenges in crop disease identification or precision agriculture.

Reply: Thank you for your suggestion. In the introduction, we have addressed the current challenges in disease classification, including the complexity of crop disease characteristics, the difficulty of integrating CNNs with ViTs, and the unsuitability of ViTs for resource-constrained devices. We also explained in the methods and results sections why our model can effectively tackle these challenges.

Reviewer 1

1. Comment: The most important pre-processing techniques for agricultural images are not shown.

Reply: We have changed the title of section 2.1 from “Datasets” to “Datasets and preprocessing”, and expanded this section to include explanations of image preprocessing techniques, encompassing dataset partitioning, image cropping, image normalization. In addition, we have described the data augmentation techniques used in the training process in section 3.1.

2. Comment: Add additional research to review the previous research review for the research field.

Reply: We have reviewed applied research on combining CNNs with transformers for crop disease identification and reviewed their shortcomings. Also, we added references to the following literature:

Reference 24: Xie W, Zhao M, Liu Y, Yang D, Huang K, Fan C, et al. Recent advances in Transformer technology for agriculture: A comprehensive survey. Eng Appl Artif Intell. 2024;138: 109412. doi:10.1016/j.engappai.2024.109412.

Reference 26: Yu S, Xie L, Huang Q. Inception convolutional vision transformers for plant disease identi-fication. Internet Things. 2023;21: 100650. doi:10.1016/j.iot.2022.100650.

Reference 27: Wang Y, Chen Y, Wang D. Convolution Network Enlightened Transformer for Regional Crop Disease Classification. Electronics. 2022;11: 3174. doi:10.3390/electronics11193174.

3. Comment: Please explain figure 7 better.

Reply: We apologize for any confusion caused by Figure 7. We have explained the calculation process of the ALP-FFN module in more detail and added the calculation formulas. In addition, we noticed that figure 7 is poorly drawn, so we have redrawn figure 7 to make it more understandable.

4. Comment: Discuss the results listed in the tables and the criteria that were adopted to establish a clear vision.

Reply: We have improved the discussion of the experimental results in sections 3.2.1 and 3.2.2, and added new discussion in comparison with previous work. In addition, the explanation of the performance metrics has been given in section 3.2.

5. Comment: Clarification of proposed future work from this manuscript.

Reply: In the conclusion section, we have outlined our future work plan as follows:

(1) Collecting datasets from real field environments.

(2) Expanding the model to include more crop and disease types.

Reviewer 2

1. Comment: Your figures appears as an appendix, fix them within the papers even if it takes a whole page.

Reply: Thank you for your advice. However, the journal requires that figures cannot be included in the main manuscript file. Each figure must be prepared and submitted as a separate document.

2. Comment: Your tables and format is not well arranged, make your tables in a page, not in separate pages.

Reply: We have redesigned and adjusted the format and layout of all the tables to ensure that each table is fully displayed on its own page.

3. Comment: Follow the actual format.

Reply: We have carefully revised the manuscript format to meet the journal requirements.

Thank you again for your positive and constructive comments and suggestions on our manuscript.

We hope you will find our revised manuscript acceptable for publication.

---

## [Decision Letter · Decision Letter 1]

11 Mar 2025

A dual-branch model combining convolution and Vision Transformer for crop disease classification

PONE-D-24-46420R1

Dear Dr. Zhang,

We’re pleased to inform you that your manuscript has been judged scientifically suitable for publication and will be formally accepted for publication once it meets all outstanding technical requirements.

Kind regards,

Venkata Krishna Parimala

Academic Editor

PLOS ONE

Additional Editor Comments (optional):

Reviewers' comments:

Reviewer's Responses to Questions

**Comments to the Author**

1. If the authors have adequately addressed your comments raised in a previous round of review and you feel that this manuscript is now acceptable for publication, you may indicate that here to bypass the “Comments to the Author” section, enter your conflict of interest statement in the “Confidential to Editor” section, and submit your "Accept" recommendation.

Reviewer #1: (No Response)

Reviewer #2: All comments have been addressed

2. Is the manuscript technically sound, and do the data support the conclusions?

Reviewer #1: (No Response)

Reviewer #2: Yes

3. Has the statistical analysis been performed appropriately and rigorously? 

Reviewer #1: (No Response)

Reviewer #2: Yes

4. Have the authors made all data underlying the findings in their manuscript fully available?

Reviewer #1: (No Response)

Reviewer #2: Yes

5. Is the manuscript presented in an intelligible fashion and written in standard English?

Reviewer #1: (No Response)

Reviewer #2: Yes

6. Review Comments to the Author

Reviewer #1: (No Response)

Reviewer #2: 1.Expand the Comparison:

Include comparisons with more recent models, especially those that also combine CNNs and Transformers, to provide a more comprehensive evaluation of the proposed model's performance.

2. Computational Efficiency Analysis:

Provide a detailed analysis of the model's computational efficiency, including inference time and energy consumption, to demonstrate its suitability for deployment on edge devices.

3. Ethical Considerations:

Add a brief discussion on ethical considerations related to the use of AI in agriculture, including potential biases in the dataset and the impact of misclassification on farmers.

4. Hyperparameter Tuning:

Include a discussion on hyperparameter sensitivity and tuning to provide insights into the robustness of the proposed model.

5. Broader Impact Discussion:

Expand the discussion on the potential impact of the proposed model on precision agriculture, including how it could be integrated into existing agricultural practices and its potential to reduce crop losses due to diseases

7. PLOS authors have the option to publish the peer review history of their article (what does this mean? ). If published, this will include your full peer review and any attached files.

**Do you want your identity to be public for this peer review?** For information about this choice, including consent withdrawal, please see our Privacy Policy .

Reviewer #1: No

Reviewer #2: No

---

## [Editor Report · Acceptance letter]

PONE-D-24-46420R1

PLOS ONE

Dear Dr. Zhang,

I'm pleased to inform you that your manuscript has been deemed suitable for publication in PLOS ONE. Congratulations! Your manuscript is now being handed over to our production team.

Kind regards,

on behalf of

Professor Venkata Krishna Parimala

Academic Editor

PLOS ONE